# Nanotechnology Involved in Treating Urinary Tract Infections: An Overview

**DOI:** 10.3390/nano13030555

**Published:** 2023-01-30

**Authors:** Andreea Crintea, Rahela Carpa, Andrei-Otto Mitre, Robert Istvan Petho, Vlad-Florin Chelaru, Sebastian-Mihail Nădășan, Lidia Neamti, Alina Gabriela Dutu

**Affiliations:** 1Department of Medical Biochemistry, Faculty of Medicine, Iuliu Hatieganu University of Medicine and Pharmacy, 400349 Cluj-Napoca, Romania; 2Department of Molecular Biology and Biotechnology, Faculty of Biology and Geology, Babes-Bolyai University, 400084 Cluj-Napoca, Romania; 3Department of Pathophysiology, Faculty of Medicine, Iuliu Hatieganu University of Medicine and Pharmacy, 400349 Cluj-Napoca, Romania; 4Faculty of Medicine, Iuliu Hatieganu University of Medicine and Pharmacy, 400349 Cluj-Napoca, Romania

**Keywords:** urinary tract infections, antimicrobial treatment, nanotechnology, organic nanoparticles, inorganic nanoparticles, mixt nanoparticles, biocompatibility

## Abstract

Considered as the most frequent contaminations that do not require hospitalization, urinary tract infections (UTIs) are largely known to cause significant personal burdens on patients. Although UTIs overall are highly preventable health issues, the recourse to antibiotics as drug treatments for these infections is a worryingly spread approach that should be addressed and gradually overcome in a contemporary, modernized healthcare system. With a virtually alarming global rise of antibiotic resistance overall, nanotechnologies may prove to be the much-needed ‘lifebuoy’ that will eventually suppress this prejudicial phenomenon. This review aims to present the most promising, currently known nano-solutions, with glimpses on clinical and epidemiological aspects of the UTIs, prospective diagnostic instruments, and non-antibiotic treatments, all of these engulfed in a comprehensive overview.

## 1. Introduction

Except for the urethra, the urinary tract remains uncolonized, although occasional contamination of bladder urine is possible, especially in women due to the short and direct trajectory of the urethra [1,2]. In order of importance, the antimicrobial barriers of the urinary tract are: continuity of the uroepithelium doubled by mucopolysaccharides, a glycosaminoglycan that antagonizes the adhesion of bacteria lacking special ligands, washing of the mucosa through the normal urine flow (with a minimum dead space by complete evacuation of the bladder during normal urination), high osmolarity and high concentration of urea as well as a low pH value of urine, and the antibacterial effects of prostate secretion and periurethral glands [3,4,5,6,7].

Working synergistically, all these factors mentioned above create an unfavorable environment for the development of many bacteria in the urinary tract. However, like in the proximal parts of other orifices, the distal urethra is normally colonized with bacteria from the perineal skin, colon, and vulva. Thus, in the distal urethra, in moderate amounts, coagulase-negative staphylococci, Enterobacteriaceae, enterococci, *Fusobacterium* spp., Gram-positive and Gram-negative anaerobic cocci, lactobacilli, or even *Candida* species are very common [8]. Eventually, this resident microbiota also plays a discrete role in maintaining a proper barrier against threatening pathogens, establishing both fine and fascinating lines between what we can call ‘bad’ and ‘good’ germs [9].

Urinary tract infections (UTI) represent an enormous health problem globally, affecting both sexes, regardless of age, and they can vary in symptomatology and prognosis, being complicated or uncomplicated [10].

The diagnosis of UTI is done based on the reporting of symptoms that are supported by laboratory tests represented by urinalysis and urine culture [10]. The microbiological diagnosis of urinary infections is made by isolating pathogenic organisms, for a positive diagnosis of UTI [11]. The antibiogram helps to check the effectiveness of the therapy in fighting the infection. Urine culture is not mandatory in patients who do not present risk factors for developing a complicated UTI, but with its help the possible development of antibiotic resistance of uropathogens is reduced [10].

Catheter-associated urinary tract infections have significant clinical and economic consequences. Catheter-associated bacteriuria may be associated with excess mortality, even after controlling for underlying factors such as severity of illness and comorbidities; hospital-onset bloodstream infection resulting from a urinary source has a case fatality of 32.8% [12].

European studies found that 15–25% of patients in hospitals and 5% of those in elderly homes have been subjected to urinary catheter insertion. Further studies estimate that 41–58% of existing catheters are most likely needless. The risk to acquire CAUTI increases by 5% with daily catheter use. The CAUTI annual cost in the UK is over EUR 100 million annually (per episode it is over EUR 2000) (European Centre for Disease Control (ECDC) and European Network for Safer Healthcare (ENSH)).

Data from ECDC and ENSH help us to know that 6.5% of hospitalized European patients suffered a healthcare-associated infection and about 19% to 20% had a UTI. Urinary tract infections are one of the three most common medical infections. Catheter-associated urinary tract infections also mean an additional EUR 390 million in annual costs for the EU, and also around 400,000 additional hospital days, equaling around 3800 life years.

With around 400,000 hospitalizations annually just in the United States and a burdening cost of around USD 2.8 billion for the healthcare system [13], it is clear that UTIs could be managed through more refined approaches, considering their current large spread and financial footprint. The notably increasing incidence of UTIs by 52% from 1998 to 2011 [13], together with a doubled cost per patient care in the same period [14], outlines the above-mentioned statement. Another interesting study revealed that EUR 13.5 million was spent only in France in 2012 for the treatment of suspected UTIs that eventually had negative cultures of urine [15]. This analysis strongly suggests that the diagnostic tools should also be considered for improvement and as a cost-cutting instrument. Finally, the rising threat of antibiotic resistance among infection cases in general and UTIs in particular has also led to somber predictions in terms of cost risings. A study from Lebanon shown that patients infected with resistant urobacteria have 29% higher hospitalization costs, not only because the aggravated forms of their conditions, but also because of a longer length of stay in the hospital [16]. In the USA, the healthcare system may be under even more pressure due to this phenomenon, since it may add USD 2.2 billion annually to the costs with UTI patients, which is roughly the total amount of money spent on the remedy of all UTI cases [17].

To sum up, there are many aspects of this type of infection that may influence not only their clinical outcome, but, implicitly, the cost involved in them, with the most significant one being represented by the bacteria’s resistance to antibiotics. Considering the almost ubiquitous incidence of UTIs and the already-known economic burden generated by this condition for the national health services, it is our conviction that more experimental studies and reviews addressing novel approaches in the management of these conditions should be encouraged, published, and highlighted.

## 2. Clinical-Epidemiological Considerations

The urinary tract infection may be inapparent or overt. It affects any segment of the urinary tract and, due to the anatomical relationships, once installed in one segment, it extends to the others, leading to related diseases: urethritis, prostatitis, cystitis, and pyelonephritis [18]. In view of its manifestations, they generally range from episodic, limited acute episodes to recurrent, chronic conditions. Regarding its localization, kidney limitation can be a constant stage, as in leptospirosis, or occasional, such as in pyelonephritis with *Candida albicans*, *Haemophilus influenzae*, staphylococcal renal abscesses, renal tuberculosis, and so on. On the other side, renal elimination of the bacterium is achieved without clinically manifested kidney damage [19]. Finally, it is worth being mentioned that neither serum antibodies (IgM, IgG) nor those excreted in the urine (IgG, IgA) seem to play a role in kidney defense. In inferior urinary tract infections, the immune response is absent, and the infection is largely superficial [20].

Regarding the most affected age categories, urinary tract infections are relatively frequently diagnosed at all ages. However, due to a less efficient immune function, cumulative comorbidities, or increased risks related to nosocomial infections, the elderly have higher chances to develop symptomatic UTIs [21,22]. In newborns and infants, the prevalence is higher in boys than in girls. Subsequently, the ratio reverses more and more sharply until adulthood. However, the difference between sexes disappears in the elderly, due to the faster increase in the prevalence in men related to prostate diseases and the instruments they require. It is also worth mentioning that about 10–20% of women experience urinary tract infections at least once in their lifetime [23].

Even if we mentioned above several anti-infective barriers of the urinary tract, these anatomo-functional components still have various degrees of instability. For example, unlike other tissues, on the bladder mucosa, the phagocytosis of surfaces is practically absent [24,25]. Furthermore, residual bacteria can multiply in the urine film in the interval between urination, while complement effector and phagocytosis are useless in the hypertonic conditions of the renal medulla [25,26,27]. Different conditions may modify the balance of the microorganism and host to the detriment of the host, with the appearance of colonization or urinary tract infections. Obviously, the ascending path is the most important for conditioned bacterial pathogenic access in the urinary tract [28].

From a microbiological perspective, *Escherichia coli* is the most common etiological agent of urinary tract infections. Only a few serogroups of *Escherichia coli* have uropathogenic abilities, such as: recognizing sphingolipid receptors on the uroepithelium, the abundance of the K antigen, and resistance to the bactericidal action of blood serum or hemolysin production. Other serogroups, although predominant in the colon microbiota, produce only occasionally abortive urinary colonization or cystitis [29,30,31].

The rest of the Enterobacteriaceae, enterococci, or *Pseudomonas aeruginosa* cause chronic or recurrent infections; due to obstructive uropathy and neurogenic bladder dysfunction, *Staphylococcus saprophyticus* is a more common cause of cystitis in sexually active young women [32,33]. *Corynebacterium urealyticum* has recently been reported to cause urinary tract infections in patients at risk for hospitalization, prolonged antibiotic treatment, or urological interventions. *Candida albicans* causes specific infection in metabolically uncontrolled diabetics [34,35]. Urease-positive species, like *P. mirabilis* and *S. saprophyticus*, alkalize urine and promote the production of stones, which creates major difficulties in eradicating certain pyelonephritis [36].

Concerning the viral UTIs, serovar adenoviruses 11 and 21 cause hemorrhagic cystitis in children, but, in general, viruses that are excreted in the urine do not cause urinary tract infections [37].

## 3. Antibiotics

Chemotherapeutic substances are unknown products to the body which suppress or damage tumor cells and microorganisms [38,39,40]. Among them, we can mention antibiotics, antifungals, and so on. Antibiotics are substances with antibacterial effect, effective on bacteria in low concentrations, which observe the principle of selective toxicity, acting at the molecular level on an essential metabolic pathway for bacteria; the metabolic pathway being absent in the eukaryotic cell, or, if present in eukaryotes, not being affected by the antibiotic [41,42,43,44].

The modern era of antibiotics began with the discovery of penicillin by Sir Alexander Fleming in 1928 [45]. Even at the time, it was a revolutionary medicine; penicillin resistance became soon a critical issue and new antibiotics began to be discovered, developed, and implemented. Thus, from the late 1960s to the early 1980s, the pharmaceutical industry introduced new antibiotics to the market to solve the problem of resistance of various microorganisms to antibiotics [46,47,48,49].

In 2015, many decades after the first patients were treated, infections caused by microorganisms became a new threat [49,50]. Antibiotics fight bacteria by inhibiting certain vital processes of bacterial cells or bacterial metabolism [51,52] and can target the bacterial wall or cytoplasmic membrane or localized processes in the bacterial cytoplasm such as protein synthesis, DNA replication, or both [51,53,54]. For each of the groups listed above, the classification into families is based on the chemical structure of the various substances, with each family having a specific molecular mechanism [55,56].

Antibiotic resistance is an important and topical issue, which refers to the situation where antibiotics that usually kill bacteria no longer do so [55]. The study of antibiotic resistance has focused especially on the consequences of resistant microorganisms, as it can interfere with the treatment of infections [57,58]. Thus, it was demonstrated that bacteria were able to develop resistance, not only because the acquisition of antibiotic-resistant genes, but also as a result of excessive and abusive use of drugs and the lack of new drugs [49,59]. In consequence, patients infected with resistant bacteria will manifest symptoms for a longer time, and the chances of the conditions getting worse will be higher. Moreover, epidemics will be on the rise and more and more people will be at risk of infection [60,61,62,63].

The main causes of antibiotic resistance are their incorrect use: either empirical treatment is performed (without performing an antibiogram to test germ sensitivity); strong antibiotics are prescribed for infections that could be treated with simple antibiotics; or administration is in too small amounts, for too short a period, or at too long intervals [59,64]. The most worrying cases are for the species of the family Enterobacteriaceae—*Escherichia coli* and the *Klebsiella* genus.

Resistance of microorganisms to antibiotics can be natural or acquired. The natural resistance of a species or genus is a characteristic of its own [65,66,67]. It is always transmissible to offspring because it is carried by the chromosome. Thus, natural resistance determines the wild phenotypes of bacterial species against antibiotics [68,69]. Unlike natural resistance, acquired resistance refers only to a proportion of strains belonging to a species or genus, variable over time. Acquired resistance exists through the accumulation of one or more resistance mechanisms that determine a certain resistance phenotype, a different phenotype from the wild one [70,71,72]. Among the mechanisms for obtaining antibiotic resistance, we mention the genetic mechanisms for acquiring antibiotic resistance and the biochemical mechanisms. As seen in Figure 1, the change in the permeability of the bacteria to antibiotics occurs through mutation [67,73].

Antibiotics enter the Gram-negative bacteria through the pores or lipopolysaccharide of the outer membrane [74]. Changing the permeability of bacteria by mutation, by changing porins or lipopolysaccharides, prevents the penetration of antibiotics into the bacteria [75,76,77]. By changing the permeability of bacteria to hydrophilic or hydrophobic antibiotics, co-resistance or cross-resistance is installed against a family of antibiotics, a group of antibiotics with the same physicochemical properties [74,78]. By changing the permeability, resistance to various families of antibiotics can be acquired: beta-lactams, tetracyclines, fennel, polymyxins, and so on [79,80,81]. Modification of antibiotic transport in bacteria occurs by mutation, meaning that the antibiotic can no longer be transported in bacteria. Other biochemical mechanisms for acquiring antibiotic resistance that may be mentioned are the modification of the antibiotic action target, the substitution of the action target, the expulsion of the antibiotic, or the inactivation/modification of the antibiotic by inactivation enzymes (Figure 2) [82,83,84]. From the last category mentioned above, it can be said that the inactivation or modification enzymes of some antibiotics generally have a plasmid and rarely chromosomal determinism and can act either in the extracellular or intracellular periplasmic space of bacteria (beta-lactamases in Gram-negative bacteria) [70].

Finally, a prediction generated in 2016 estimated that more than 10 million people will die annually by 2050 because of the antimicrobial resistance (AMR) phenomenon [85]. Corroborating the fact that UTIs account for a great percent of all these infections and the fact that, recently, there has been a clearly rising AMR trend, especially among these kinds of infections [86,87] outlines the need for novel and more efficient clinical approaches regarding the management of UTIs. With an inefficiency already varying from 32 to 60% in terms of antibiotic medications around the globe [86], it seems that classical therapy is more and more suboptimal for the clearance of infections.

With these brief statistics in mind, we chose to present a complete picture on how the antimicrobial medication evolved, rather than just enumerating novel technologies used for urinary infection healing. Starting from basic, primary therapeutics and expanding to the current state of research on UTIs treatments, in the following chapter we extensively detail the nanotechnology’s involvement and their perspectives in this clinical topic.

## 4. Nanotechnology Used as a Diagnostic Method and in Antimicrobial Treatment

Diseases caused by various pathogens are numerous and can be easily transmitted. Rapid and specific determination of pathogens is particularly important to determine the source of infection and the spread of infection but also the proper treatment of patients [88]. The complexity and variety of pathogens, and also the incubation period until the onset of the first symptoms of the pathology, can make the diagnosis difficult. Modern diagnostic methods such as ELISA and PCR are characterized by high sensitivity and reproducibility. These methods also have disadvantages: laborious preparation of samples and long time until conclusive results are obtained [89].

Nanotechnology presents a great opportunity for the development of fast techniques, characterized by high sensitivity, specificity, and low costs for the detection of infections caused by various pathogens (Figure 3) [90,91,92].

Various types of nanomaterials, such as magnetic and fluorescent gold nanoparticles, are used to determine microorganisms. The versatility of these inorganic nanoparticles, together with the interactions with different target molecules or pathogens, makes their use promising for diagnosis [92,93,94,95]. It should be noted that these diagnostic strategies, which are based on nanoparticles, depend on the recognition of different sequences in the bacterial genome and that is why no new strains can be identified [96]. Resistant strains continue to emerge, and nanotechnology needs to provide new ways to identify both the presence of pathogens and the sensitivity of pathogens to antibiotics [97]. There is great interest in the rapid development of a diagnostic method for urinary tract infections. As early as 1980, various methods began to be developed to help detect urinary tract infections—for example, immersing a strip in urine to assess the esterase produced by leukocytes and nitrites [98]. More recently, automated analyzers that are based on the principle of flow cytometry allow rapid detection of bacteria, leukocytes, erythrocytes, epithelial cells, crystals, and so on. Although this method is quite fast, it cannot provide the microbiological diagnosis nor the susceptibility of microorganisms to different classes of antibiotics [99]. The standard procedure for detecting urinary tract infections is based on several steps that include the collection and transport of the biological sample. Once the results are available, each clinical laboratory has to notify the clinician about the result of the analysis, who in turn informs the patient about the result obtained [100,101]. All these steps that need to be taken are time-consuming and can delay the provision of effective treatment. The new generation of biosensors based on micro- and nanotechnologies offers the possibility of a very efficient molecular diagnosis [102,103]. A biosensor is a device consisting of the combination of two main elements: a sensitive, biological element, the bioreceptor, which, immobilized on a support, can specifically recognize a target substance present in a complex environment; and a physicochemical system, which captures the changes in the bioreceptor in contact with the target substance and translates them into a useful electrical signal [104,105]. The bioreceptor can be of the catalytic (metabolic) type, such as an enzyme, a cell organelle, a microorganism, or even a tissue, but also of the non-catalytic (affinity) type, such as an antigen or an antibody [106,107]. In developing a biosensor for urinary tract infections, several aspects need to be considered. To perform the analyses of interest in the case of urinary tract infections, several steps are required: sample preparation—pipetting, centrifugation, and washing. The sensitivity of the biosensor can be determined by several factors, such as matrix effects and nonspecific binding [100,108]. A urinary tract infection sensor should meet several criteria: the ability to detect infection, quick performance of the analysis, the automatic preparation of the sample with minimal technician intervention, the ability to identify pathogens, and antimicrobial sensitivity, so as to be versatile enough to be able to identify different pathogens in different infections [100]. The emergence of antibiotic resistance of microorganisms has become a threat to human health. The study of antibiotic resistance has focused on pathogenic microorganisms, but also on the consequences that resistant microorganisms have on health [58,109]. The development of antibiotic resistance is very important from a clinical point of view, as it can be detrimental to the treatment of infections. Shortly after the introduction of antibiotics for human therapy, bacteria were able to develop resistance, as a consequence of mutations, the acquisition of antibiotic-resistant genes, and the overuse and abuse of drugs, in addition to the lack of new drugs [59]. The natural resistance of a species or genus is a characteristic of its own. It is always transmissible to offspring because it is carried by the chromosome. Thus, natural resistance determines the wild phenotypes of bacterial species against antibiotics. Unlike natural resistance, acquired resistance refers only to a proportion of strains belonging to a species or genus, variable over time [110]. Acquired resistance exists through the accumulation of one or more resistance mechanisms that determine a certain resistance phenotype, a different phenotype from the wild one [111]. There are three categories of mechanisms responsible for the acquired resistance of bacteria to antibiotics. The first one is represented by a reduction in the amount of antibiotic that reaches the target by decreasing permeability or by the appearance of efflux systems [112]. Decreased permeability is found especially in Gram-negative bacteria, in which the pores are partially or completely blocked or may disappear. The crossing can also be slowed down due to mutations, so that the accessibility of antibiotics to the pores outside the outer membrane decreases, and using outflow systems that use a proton-motor force that expels the antibiotic from the moment it enters the bacterial cell [113,114].

Other different mechanisms responsible for the acquired resistance of bacteria to antibiotics are represented by the modification of the antibiotic target, which can be achieved by mutations in the genes encoding the antibiotic target, by acquiring foreign genes, or by the inactivation of the antibiotic, which is the most common mechanism encountered in infectious pathology [73,112]. Of the resistance mechanisms, one of the important ones is the expression of enzymes that degrade or modify antibiotics, such as β-lactamases, which remains one of the significant threats against antibiotic efficiency [42].

Nanomedicine plays an important role in enhancing the effectiveness of existing therapies through increasing the stability and the physicochemical properties of antibiotics and antibiotic release and offering an opportunity for biofilm internalization, further increasing the capability for target delivery of drugs to the infection site [115,116,117]. It was also found that, in addition to improving the therapeutic activity of antibacterial agents, nano-sized systems also restrain the stimulation of resistance through overcoming the developed resistance strategies of the bacteria [118,119]. These strategies involve drug decomposition by β-lactamase or efflux pumps or modification of the bacterial cell wall. As most types of nanoparticles can overcome at least one of these common resistance mechanisms, an increased number of nanoparticle variants have been used as an improved line of defense against antibacterial resistance [120].

### 4.1. Organic Nanoparticle Therapy Approaches

The overall advantages lately brought by the usage of organic nanoparticles are already well known. Starting from a more precise targeting of various tissues/environments or even cells and going to a strictly controlled on-site dosage and enhanced colloidal stability of small drugs (like proteins or nucleic acids), these nano-sized carriers revolutionized the concept of modern therapy (Figure 4) [121,122].

In the case of catheters and other traditional medical devices/approaches used in treating urinary diseases, a great risk for the patient is that of developing biofilms (Figure 5) [123,124].

In the light of this issue, we present several promising findings related to the usage of nanoparticles in UTIs therapy management.

The first example refers to a food sweetener naturally present in ginger called zingerone [125]. Zingerone nanoparticles (ZNPs) were tested in vivo on mice as an alternative to standard therapies against *Pseudomonas aeruginosa*, one of the bacteria able to produce biofilms [126]. After treatment, both mice were treated with zingerone and with ZNPs, showing a decreased bacterial count in both renal and bladder tissues compared to untreated controls. Moreover, while controls showed increases in some inflammatory markers (MDA, MPO, and reactive nitrogen intermediates), mice treated with zingerone or ZNPs showed decreases in those markers, ZNPs having a stronger effect than zingerone itself. Lastly, ZNPs increased serum sensitivity and phagocytosis of *Pseudomonas aeruginosa* compared to zingerone, which summarizes perfectly the importance of considering nanoparticle formulations rather than classical, simple drugs for the treatment of UTIs [127,128].

Nanocarriers were also considered as remedies for UTIs in the light of improved bioavailability and better retention in the target tissue. For example, Brauner et al. [129] proposed a formulation of Trimethoprim (TMP) as poly(D,L-lactic-coglycolic acid) (PLGA)-based nanospheres, further conditioned with wheat germ agglutinin (WGA), to be administered intravesically through a catheter [129,130,131]. TMP is used to treat uropathogenic *Escherichia coli* infections, and while usually TMP is administered orally, there have been other cases of medicine administered intravesically because of the increased local concentration and to avoid absorption losses or systemic effects [132]. Moreover, WGA is used to mimic the role of *Escherichia coli* type 1 fimbriae (FimH), both having increased binding activity for the bladder epithelial tissue [133]. PLGA-coated TMP nanospheres, with and without WGA added to the surface, were tested on immortalized human uroepithelial cells. On repeated washings of the urothelium after the initial application of the formulations, the authors noted higher adhesions for nanoparticles of higher WGA density. Overall, there was no visible connection between TMP loading and nanoparticles adhesion. Lower pH levels and higher incubation times favored the adhesion of the nanoparticles to the urothelium [129,134].

As antibiotic resistance was already identified as a major worldwide concern, solutions involving organic nanoparticles can also be used to prevent antibiotic resistance in target bacteria. For example, Amikacin (AK) resistance can be gained through multiple mechanisms (mutations of the 30S subunit of the ribosome, aminoglycoside-modifying enzymes (AME), changes in lipid content of the plasma membrane, efflux pumps) [73,135,136]. Moreover, antibiotics like Amikacin have narrow therapeutic indexes and strong systemic countereffects. In such cases, nanocarriers can be used to improve bioavailability and target specific tissues [137]. As a solution for this issue, scientists designed liposomes that can entrap a range of antibiotics (including Amikacin) which are then able to fuse with the bacteria’s membrane and deliver their content directly into the antibiotic-resistant bacteria. In this way, on the outside of the cell, the antibiotic molecule is protected from AME action by the liposome vehicle [138,139].

Additionally, if the above-mentioned strategy is still insufficient for the desired results, it is also worth mentioning that surfactants such as Poloxamine 908 are able to influence the uptake in desired tissues (decreasing the liver uptake while increasing the spleen uptake) [140]. Some polymeric nanoparticles are also able to increase uptake in a specific tissue, further helping to target the desired infection location (such as Amikacin-bearing PLGA nanoparticles that have an increased uptake in Peyer’s patches compared to plain Amikacin) [141].

Going even further, considering that the localization of UTIs may sometimes be hard to determine, Song et al. [142] created a copolymer (LGseseTAPEG) of PLGA, selenocystamine, and methoxypoly(ethylene glycol) tetraacid which is sensitive to local oxidative stress [142]. 

Besides being effective drug carriers, organic nanoparticles can also be used as detectors of pathogens. One example of a nano-sized UTI detection system is Cranberry proanthocyanidins (PAC), which can interact with fimbriae of extra-intestinal *Escherichia coli* [143]. PAC-polyaniline (PANI) nanocomposites were created and used on screen-printed electrodes (SPE) to assess electrochemical changes caused by the interaction between PAC and *Escherichia coli* [144,145]. The resulting PAC-PANI nanocomposites on SPE produced a linear electrical response towards *Escherichia coli* presence over the range of 1 to 70,000 colony-forming unit/mL (CFU/mL), and the limit of quantification was 1 CFU/mL, much lower than earlier approaches in the literature. Moreover, the electrical response was different for other pathogens, and as such the approach was also specific for *Escherichia coli* [144].

Finally, in a similar manner, nanoparticles may be used to create noninvasive, urine-based tests for diseases that are not specific to the urinary system [146]. In a clinical study, HIV-positive patients presenting neurological symptoms specific for toxoplasmic encephalitis (TE) were tested both through qPCR and through NPs of poly-N-isopropylacrylamide dyed with reactive blue-221 [147,148]. The NPs, which captured and concentrated relevant antigens from urine, were then evaluated through western blot immunoassay. Unfortunately, due to the lack of a golden standard test (qPCR has a sensitivity of under 70%), the authors were unable to compute the sensitivity and specificity of the NP-based technique described [147].

### 4.2. Inorganic Nanoparticle Therapy Approaches

As a more stable and less toxic alternative, inorganic nanoparticles may be used to respond even better to the challenges that emerge in the therapy of UTIs. Additionally, considering that many of these NPs have antibiotic properties themselves, reducing bacteriuria in infections caused by both multi-drug resistant (MDR) and common pathogens, they may be especially recommended for the treatment of UTIs [131]. Thus, we try to present the usage of inorganic NPs in UTI management in a more comprehensive manner.

In the search for effective and new ways to treat urinary tract infections caused by MDR pathogens, biogenic selenium nanoparticles (Se NPs) have been created through two different methods: by biogenic synthesis using *Penicillium chrysogenum* filtrate and by incorporating Se with Gentamicin (CN) under gamma radiation (green synthesis) [149]. The Se NPs incorporated with CN (Se NPs-CN) exhibited antimicrobial activity against *Escherichia coli*, *Staphylococcus aureus*, *Bacillus subtilis*, *Pseudomonas aeruginosa*, and also towards *Candida albicans.* The biogenic Se NPs showed effectiveness against *Staphylococcus aureus*, *Candida albicans*, and *Pseudomonas aeruginosa* [150,151,152]. The antimicrobial activity was more significant compared to the precursor sodium selenite, the fungal filtrate, Gentamicin alone, and other standard antimicrobial agents. Additionally, the Se NPs-CN presented a more efficient activity compared to the biogenic Se NPs [151,153]. The Gram-negative bacteria were more susceptible to the Se NPs action compared to the Gram-positive. It was also observed that Se NPs-CN had antibiofilm activity against *Escherichia coli*, *Pseudomonas aeruginosa*, and *Staphylococcus aureus* [154,155]. The mechanism of the Se NPs is based on the fact that selenium is capable of producing highly effective superoxide radicals which create a thiol deficiency in the bacteria, which causes cell death (Figure 6). It is also important to note that a dose higher than 1.8 mg/kg produced chronic hazardous effects on tested mice [151,156].

Since metal NPs do not aim to induce alterations in metabolic pathways, they cannot trigger the bacterial resistance mechanisms, so these agents can be considered excellent candidates for the treatment of UTIs. Still, the production process of these instruments may have detrimental effects on the environment and raise questions with regard to the side effects manifested in the patients [36,131,157,158]. To overcome these concerns, M. Abd Elkodous et al. [159] studied zinc oxide nanoparticles (ZnO NPs) produced in an eco-friendly and cost-effective manner through the sol-gel method [159]. The nanoparticles were tested against multidrug-resistant bacteria involved in UTIs. The ZnO NPs presented high thermal and chemical stability, but also anti-corrosive properties. The manufacturing process (the sol-gel method) is quite simple and it provides pure particles [160,161]. The synthesized ZnO NPs showed antimicrobial activity against multiple UTI pathogens. They was most effective against *Bacillus subtilis*, but also effective against *Pseudomonas aeruginosa* and fungi such as *Candida tropicalis*. The ZnO NPs presented much higher antimicrobial activity than zinc nitrate or other antimicrobial agents [162,163,164]. It was also noticed that the Gram-negative bacteria are more sensitive to the ZnO NPs than the Gram-positive ones. The ZnO NPs also possess very good surface area, low crystal size and good porosity and have an anisotropic nature; all these characteristics allow the ZnO NPs to be more effective. The antibiofilm activity was presented against *Bacillus subtilis*, *Pseudomonas aeruginosa*, and *Candida tropicalis.* The ZnO NPs inhibit the formation of biofilm in the initial stage, also named the irreversible adhesion stage. The ZnO NPs hinder the production of exopolysaccharides, which in the end leads to the inability of *Bacillus subtilis* to form the biofilm [165,166]. It was also observed that the ZnO NPs had some antitumor properties; more precisely, there was an in vitro cytotoxic effect against Ehrlich ascites carcinoma (EAC). The ZnO NPs’ antioxidant potential was also confirmed through the continual reduction of zinc ions. The ZnO NPs were proven to be effective in the treatment of UTIs and fungi infections and, even further, showed great antitumor action [167,168]. Due to these particular advantages, it would be worthwhile to pursue further research of its properties and applications in industries such as food processing, pharmaceuticals, and cosmetics, in which the ZnO NPs could have a positive impact [160,169,170].

In an attempt to find new ways to diagnose UTIs in the early stages, S. Vasudevan et al. developed a photoluminescence biosensor based on ZnO Cysteamine functionalized nanoparticles [171]. This type of sensor has a fast response time, high sensitivity, and specificity. The ZnO NPs combined with Cysteamine (ZnO-Cys) have shown maximum sensitivity towards detection of N-Acyl-Homoserine Lactones (AHL) produced by Gram-negative bacteria, such as *Pseudomonas aeruginosa.* The AHL is used by the Gram-negative bacteria as a signaling molecule for intra-species and also inter-species communication [172,173]. For example, *Escherichia coli* can sense and detect AHL produced by other bacteria, which leads to the regulation of virulence of the pathogen. The ZnO NPs have photoluminescence properties and, with the help of Cysteamine, the nanoparticles can be used in binding and detecting AHL molecules [174,175]. The specificity of the nanoparticles was demonstrated by using an artificial urine medium (AUM) which contained the necessary components to be as similar as possible to human urine. *Pseudomonas aeruginosa* was also added to the AUM, and, in this way, it was possible to have a real-time detection. The ZnO-Cys detected AHL molecules after 30 min and reached a maximum of detection at 90 min, after which point there was a decrease in intensity [173].

Another study performed by Muhammad Ali Syed et al. [176] manifested a particular interest in tungsten nanoparticles [131,177]. As UTIs are one of the most frequent nosocomial infections, with most of them being caused by improper catheter usage, the cited experiment comes in response to this well-known origin of these infections. As multiple drug-resistant (MDR) pathogens, especially *Escherichia coli*, are responsible for many of the catheter-associated urinary tract infections (CAUTIs), we will outline the importance of such studies in the context of CAUTI therapy management [178,179]. The Tungsten nanoparticles (W-NPs) were tested against the MDR *Escherichia coli* isolated from hospitalized patients and its efficiency was compared to Cefotaxime sodium and Sulfamethoxazole. The *Escherichia coli* isolated in this study were resistant to both antibiotics and also presented different genes which encoded adhesins and toxins. The W-NPs, when compared to the antibiotics, showed a higher efficiency at a lower MIC (minimum inhibitory concentration) and it was clear that the W-NPs determined growth inhibition, most probably because of the accumulation of intracellular reactive oxygen species [131,176]. One of the solutions for CAUTI would be the incorporation of antimicrobial agents such as W-NPs in the catheter itself. Due to the W-NPs’ nonspecific way of action, the risk of developing bacterial resistance is much lower and its design could stop the production of biofilm. Another advantage of the W-NPs in comparison to silver, for example, is the lower toxicity [180,181]. However, these nanoparticles were rather poorly studied in the context of CAUTI treatment alternatives, since there are no other relevant mentions in the literature on this issue. This fact highlights the significant perspectives posed by the development, testing, and implementation of therapies based on W-NPs [131,182].

Ureteral stent-related urinary tract infections are another issue that may be solved through the use of inorganic nanoparticles. The rate of colonization on the ureteral stents can be between 42–90%, depending on the study [183]. Not only is there a greater risk of developing a UTI, but the extraction of the stent is painful and invasive. Having that in mind, Gao et al. [184] tried to create a ureteral stent with antibacterial properties which is also biodegradable so that it does not have to be removed [184]. The stent requires flexibility and other mechanical properties to sustain the pressure of the peristaltic contractions. The stent developed by Gao et al. [184] is made out of a hyperbranched poly (amide-amine)-capped Ag shell and Au core [184]. The Ag and Au NPs were chosen because of their effective antibacterial activity and also because of the very small amount of Ag released and negative cytotoxicity. With the help of the stent’s self-cleaning capabilities, the contact surface implanted with Ag@Au NPs is constantly renewed and the debris is also removed [185,186]. In vitro tests showed that the Ag@Au NP-embedded ureteral stents were effective against *Escherichia coli* (growth inhibition over 90%) and *Sthapylococcus aureus* (growth inhibition of 99.99%). With regard to the most important mechanisms of Ag@Au NPs, it is worth mentioning that the formation of reactive oxygen species and the interaction of Ag with enzymes and proteins in the bacteria are the most important ones, both leading to cell death. In vivo experiments confirmed its degradation properties, but also the fact that it was effective in preventing UTIs, supporting the promising perspectives of this medical device [29,184,187,188].

Furthermore, as the number of UTIs caused by MDR *Escherichia coli* is on the rise because of the massive use of broad-spectrum cephalosporins, another topic related to the use of nanoparticles in urinary tract infections was studied by Sajjad et al. [189]. Nowadays, there are strains of *Escherichia coli* which produce ESBL enzymes, a fact that leads to a significant drop in antibiotics’ effectiveness. These strains presented 100% resistance to Ceftriaxone, 94% against Ciprofloxacin, and 79% against Co-trimoxazole, so the team tried to find a solution for the MDR *Escherichia coli* [190,191,192]. Silver oxide (AgO2) nanoparticles have shown potential in treating such infections, so they were considered as the main candidates for this purpose. During the production of AgO2-NPs, silver foil was used as a cheaper precursor. The study showed that there is a positive synergy between Ceftriaxone (CRO) and AgO2-NPs, as this corroboration led to a significant increase in the size of the inhibition zone against MDR *Escherichia coli.* The antibacterial mechanism is based on an increase in membrane permeability and alteration of membrane transport. Thus, not only would this treatment be more effective, but it would also be less toxic, due to the lower dose of antibiotics [189].

Besides the wide range of therapeutic-oriented usages presented above, inorganic NPs can also be considered as valuable diagnostic tools. In this regard, Vaezi et al. [193] showed in a recent study the possibilities of using an effective and inexpensive NP-based diagnosis method for urinary tract infections [193]. Other studies have shown the application of different metal quantum clusters, which have fluorescent properties. Human hemoglobin contains iron quantum clusters (Hb-FeQCs) which can be used for bioimaging. The Hb-FeQCs can bind very easily to Cu^2+^ ions and that leads to turn-off of the fluorescence of the nanostructure. *Escherichia coli* possesses enzymes that help the bacteria bind and reduce the Cu^2+^. That is why, when *Escherichia coli* is present, it will remove the Cu ions from Hb-FeQCs, and in this way the metal quantum will regain its fluorescent property in 30 min. Therefore, it would be an easy method of identifying *Escherichia coli* strains in a rapid and effective manner even at a low *Escherichia coli* concentration of only 8.3 × 103 CFU/mL. The Hb-FeQCs are also water-soluble, stable, biocompatible, and highly sensitive to bacterial concentrations, especially in the urine medium [193,194,195].

### 4.3. Mixed Nanoparticle Therapy Approaches

Bringing together the advantages of both organic and inorganic NPs while also trying to avoid the drawbacks of each category, scientists tried to develop veritable high-yielding solutions to combat UTIs [196].

For instance, Gupta et al. [197] created Au-NPs functionalized with fluoroquinolone antibiotics, resulting in up to 16-fold decreases of the minimum inhibitory concentration of the antibiotic, both on Gram-positive and Gram-negative bacteria [197]. Much in the same way, Saha et al. [198] created Au-NPs conjugated with Ampicillin (AMP), Streptomycin, and Kanamycin (KAN) and tested their efficiency and stability on *Escherichia coli*, *Micrococcus luteus*, and *Staphylococcus aureus* [198]. Kanamycin Au-NPs had a stronger inhibitory effect on *Escherichia coli* and *Staphylococcus aureus* and Streptomycin had a stronger inhibitory effect on *Micrococcus luteus*. On the other hand, *Staphylococcus aureus* was resistant to Streptomycin, and as such neither free Streptomycin nor its Au-NP form managed to inhibit bacterial growth (Table 1). Overall, functionalized NPs also had stronger activity after being exposed to heat shocks or after being stored at room temperature, compared to their classic counterparts [199,200].

Another example of NPs made from combined materials is the hybrid silver-talc nanocomposites (Ag/Tlc NCs) created by Daghian et al. [201]. These complexes were also chitosan-capped (Ag/Tlc/Csn NCs) and performed impressively not only in the cytotoxicity and antibacterial activity tests but also on wound healing assessments in mice [201]. Standard talc preparation showed weaker antibacterial activity, while Ag/Tlc NCs and Ag/Tlc/Csn NCs were both stronger than the composites containing talc, but with no significant difference between themselves. Likewise, on the antioxidant activity tests, talc had a weaker effect, while Ag/Tlc and Ag/Tlc/Csn were on par with vitamin C for the same concentration. In a total tissue bacterial count, Ag/Tlc/Csn NCs showed the lowest count (highest antibacterial activity), which was significantly different from the Ag/Tlc or Tlc formulations [202].

Last but not least, we could not ignore the emerging niche of the mobile biosensors, which, based on an organic–inorganic conglomerate, interestingly found applicability in the treatment of UTIs as well [108,203]. Punctually, using a cellulose base assembled as origami in order to aid in compiling all the necessary analytical stages, antibody-decorated gold nanoparticles (Ab-AuNPs) are used to assess the presence of *Escherichia coli* in urine samples collected from anonymized patients. When a droplet of urine containing *Escherichia coli* is dried onto one fold of the paper and then transferred onto another fold containing Ab-AuNPs, the NPs change their color to red in a dose-dependent manner. Subsequently, a mobile app was developed and calibration was performed for a mobile phone to be able to quantitatively evaluate the *Escherichia coli* concentration by capturing the image of the Ab-AuNP loaded paper. Thus, it was proven that these biosensors may be helpful in the fast and cheap point-of-care diagnosis of UTI [204].

**Table 1 nanomaterials-13-00555-t001:** Specific NPs approaches for treating UTIs.

Organics NPs	Microorganisms	Activity	Reference
KAN-chitosan NPs	*Escherichia coli*, *Proteus mirabilis*	Antibacterial	[130,131,205]
Nanodiamonds	Microorganisms	Activity	Reference
Nanodiamonds	UPEC	Antibacterial	[130,131,205]
Silver-based NPs	Microorganisms	Activity	Reference
Silver NPs	*Escherichia coli*, *Staphylococcus aureus*	Antibacterial	[130,131,205]
Silver NPs-AMP	*Enterococcus faecium*, *Staphylococcus aureus*, *Acinetobacter baumanii*, *Morganella morganii*, *Pseudomonas aeruginosa*, *Klebsiella pneumoniae*	Antibacterial	[130,131,205]
Silver NPs-AK	*Enterococcus faecium*, *Staphylococcus aureus*, *Acinetobacter baumanii*, *Morganella morganii*, *Pseudomonas aeruginosa*, *Klebsiella pneumoniae*	Antibacterial	[130,131,205]
Copper-based NPs	Microorganisms	Activity	Reference
Copper oxide NPs	MRSA, *Escherichia coli*	Antibiofilm	[130,131,205]
	UPEC	Antibacterial	[130,131,205]
Zinc-based NPs	Microorganisms	Activity	Reference
Zinc oxide NPs	Carbepenem resistant, *Acinetobacter baumanii*	Antibacterial	[130,131,205]
	UPEC	Antibiofilm	[130,131,205]
	*Candida albicans*	Antifungal	[130,131,205]
	*Candida albicans*	Antibiofilm	[130,131,205]
Gold-based NPs	Microorganisms	Activity	Reference
Gold NPs	*Escherichia coli*	Antibacterial	[130,131,205]
Gold NPs-CHX	*Klebsiella pneumoniae*	Antibiofilm	[130,131,205]

## 5. Biocompatibility

Biocompatibility refers to the ability of NPs to perform their function without causing undesirable effects. The in vivo behavior is directly connected with the NPs’ physical properties [94,206].

In order to use the inorganic NPs in the biomedical field, there are two basic requirements which have to be met: water solubility and colloidal stability. In most cases, the use of hydrophobic NPs showed a poor dispersion in biological fluids and a tendency to form aggregates. Another essential condition would be to maintain the physical properties of the NPs even after entering the organism [207,208]. The NPs’ size influences the absorption but also the clearance of the particles. NPs measuring over 100 nm are isolated by the reticuloendothelial system; meanwhile, particles which are smaller than 5.5 nm, can pass the renal glomerulus easily [209]. Another factor on which the biocompatibility of inorganic NPs depends is the surface structure. The surface structure is determined by the type and density of decoration molecules. Different molecules will confer different properties and in vivo behavior [210]. The surface charge can also modify the way a NP behaves in vivo. The charge is totally conditioned by the coating molecules, which in the end interact with the serum proteins and the cell membranes, thus influencing the biocompatibility. It has been shown that a neutral surface presented minimum interaction with the serum proteins and a prolonged blood half-life [211,212]. The biocompatibility of inorganic NPs is also strongly correlated to the nano-bio interface and all the processes which take place at its level. The result of all interactions is called the ‘biomolecule corona’, with the protein part being the most researched (protein corona) [213]. This protein corona is able to modify the adhesion to cell membranes and the physicochemical properties and the pharmacokinetics of the inorganic NPs. Stealth NPs are currently being developed for optimal action in vivo, but more research is necessary [213,214].

Since it is still largely considered an emerging field, most nanoparticle studies have been done in vitro, and, as such, tissue targeting has been only poorly studied. For greater adoption among healthcare providers and patients, nanoparticles should have the necessary characteristics to be ideally delivered by oral intake [215].

Concerning the envelope composition, chitosan is regarded as the perfect drug carrier, both due to its ability to be processed when conjugated with various medicines (including antibiotics) and for being able to increase the oral bioavailability of other medicine. For example, chitosan is fully protonated at a pH of 4, thus enabling absorption in the gastric mucosa. Absorption in slightly basic media, such as the small intestine, is managed through thiolate chitosan and other variants. Moreover, curcumin-carboxymethyl chitosan can inhibit apical P-glycoprotein drug transporter, thus improving the drug absorption. Thus, chitosan-based NPs can be used for oral intake of drugs aiming for systemic effects or specific target tissues [216,217,218].

On the other hand, as already stated, NPs have the benefit of targeting specific tissue more than systemic-administered antibiotics. Thus, nanoparticles functionalized with antibiotics might result in lesser systemic effects than classic antibiotics [157]. Pertaining to cytotoxicity, an in vitro experiment on HEK-293 human embryonic kidney cells showed that, even at a concentration of 50 μg/mL, zingerone nanoparticles were less cytotoxic than zingerone itself (93.5% vs. 89% survival rate) [128]. Mixed silver, talc, and chitosan-based NPs were safe from a cytotoxicity standpoint, in an in vivo study, for concentrations of up to 0.50 mg/mL. Interestingly, the Ag/Tlc/Csn NPs have higher toxicity than talc itself, confirming the idea that silver is responsible for cytotoxicity [219].

Lastly, NPs were proposed for use in coating medical devices, such as Foley catheters. Oftentimes, Foley catheters are used for longer periods without being replaced. In this case, they become targets of bacteria colonization and biofilm creation, in which case NPs are used to inhibit bacterial activity [220]. Chlorhexidine-loaded nanoparticles (CHX-NPs) were layered on catheters, and then those catheters were prepared as thin films of 2.5 × 2.5 cm. Hemocompatibility was assessed through hemolysis and showed no statistically significant difference between coated catheters (with and without CHX-NPs) and the uncoated catheters. Moreover, both skin irritation and sensitization tests were negative; no clinical signs of sensitivity were observed, nor was there increased presence of granulocytes or lymphocytes in skin sections [221].

## 6. Conclusions and Future Perspectives

The current review bases its structure and content on the premise that UTIs represent one of the principal reasons for antibiotic usage nowadays. As this fact was frequently encountered throughout the documentation standing at the basis of this paper, we decided to create a path through all the innovations that came as alternatives to the classic therapies, involving plain gavage of antibiotics.

Even if the nanoparticle-based therapy approaches are regarded as very recent concepts, the current review highlights the great amount of research already made in this direction, a fact that confirms the high interest manifested for the substitutes of traditional medications given for UTIs. However, we can firmly state that nano-sized therapies developed so far to treat urinary infections do respect the general tendency to consider hybrid NP-antibiotics formulations rather than simply removing classical medications. The unquestionable advantage of this strategy is that, despite almost completely disregarding decades of antibiotics research, one can potentiate the effect of the currently available drugs by using them in new formulations.

With regard to the perspectives, the example presenting the Ag/Tlc/Csn NCs is a pertinent case of how scientists should tackle the problem of NPs’ cytotoxicity [201].

Besides their multiple advantages, we mentioned that inorganic particles can have major drawbacks related to toxicity. Thus, a strategy in which various types of metals or formulations are tested in the same therapeutic complex is highly advisable.

Furthermore, the unexpected presence of mobile biosensors in this niche clinical area cannot be overlooked. Starting as a question a decade ago, the development of such devices has become factual, the already-mentioned origami-based biosensor being able to detect the presence of *Escherichia coli* in urine samples in just 7 min. As the need for proper and in-time diagnosis of UTIs becomes more and more demanding, the interest for such user-friendly and effective mobile immunosensors will certainly rise [204].

Considering all the things mentioned before, there is certainly a great amount of effort maintained and fulfilled in the conquest of finding alternative cures for UTIs in order to avoid the antibiotic resistance triggers of pathogens. Nevertheless, the reluctance of patients to be given new trial formulations, the questionable safety of these complexes, and the hazards that may be caused by their production or disposal after usage are aspects that will be solved only with gradual, careful clinical implementation and supplementary research.

## Figures and Tables

**Figure 1 nanomaterials-13-00555-f001:**
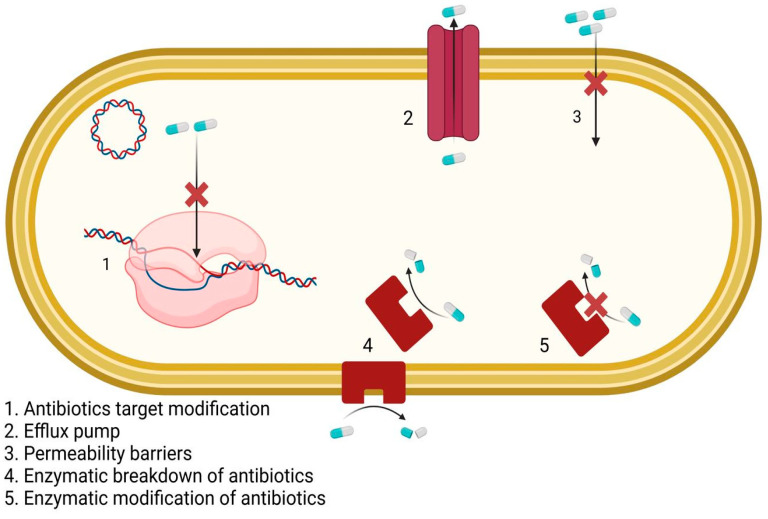
Mechanisms for obtaining antibiotic resistance.

**Figure 2 nanomaterials-13-00555-f002:**
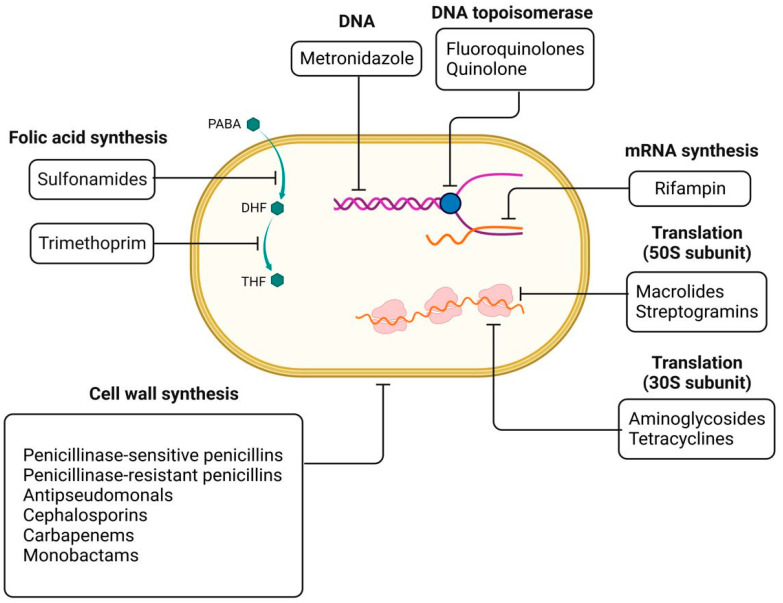
Antimicrobial therapy strategies.

**Figure 3 nanomaterials-13-00555-f003:**
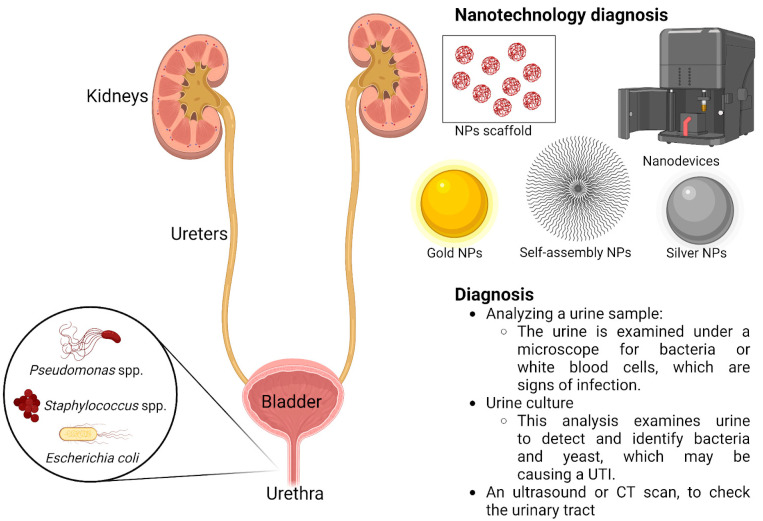
Diagnosis of urinary tract infections.

**Figure 4 nanomaterials-13-00555-f004:**
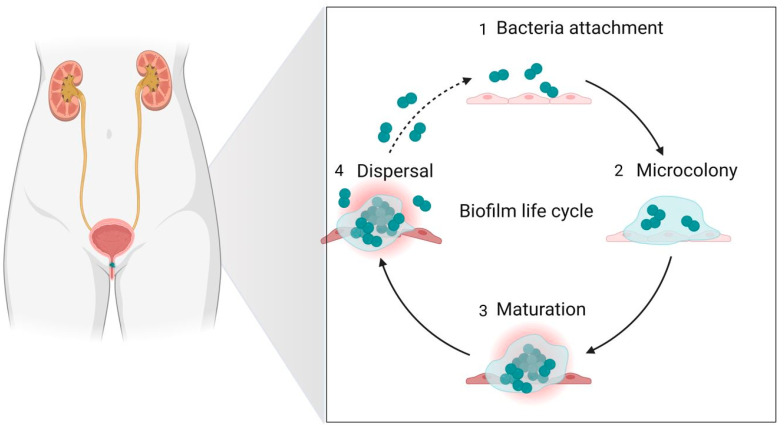
Biofilm infections in the urinary tract.

**Figure 5 nanomaterials-13-00555-f005:**
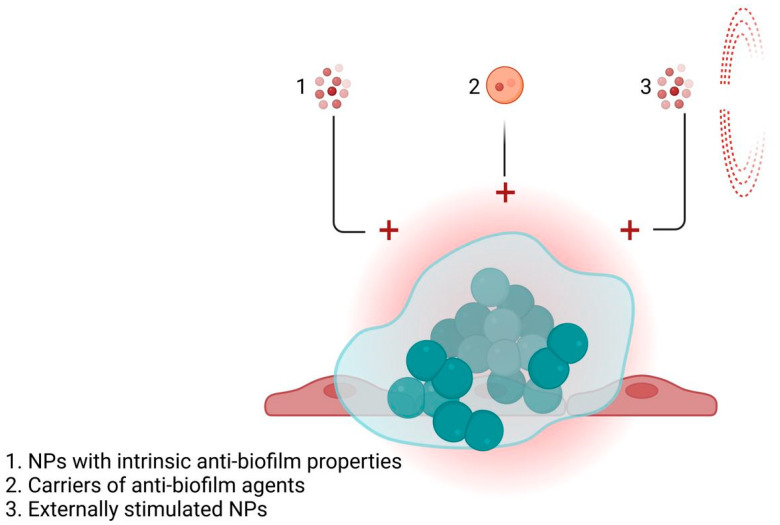
Nanoparticles for treating biofilm infections in the urinary tract.

**Figure 6 nanomaterials-13-00555-f006:**
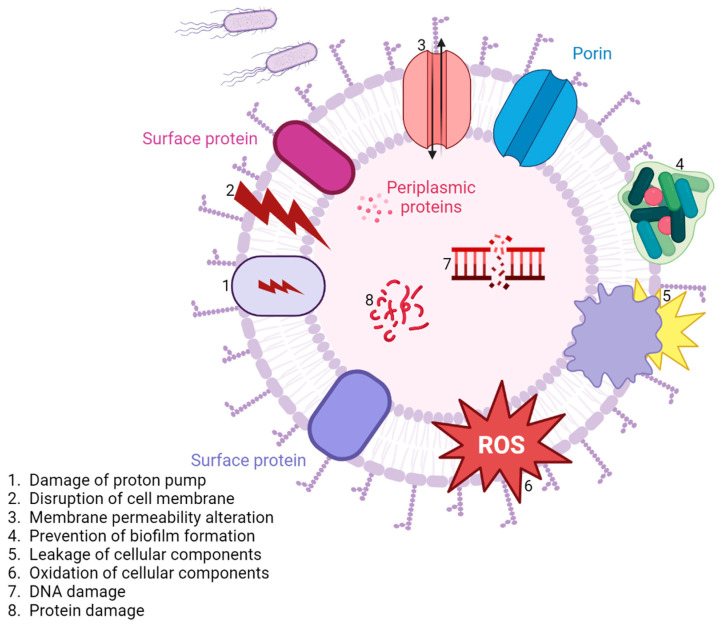
Nanoparticles—common mechanisms of action against UTIs.

## Data Availability

Not applicable.

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
