# Peer review of "Nanotechnology Involved in Treating Urinary Tract Infections: An Overview"

_nanomaterials, 2023, doi:10.3390/nano13030555_

Round 1

Reviewer 1 Report

The by Crintea et al. "Nanotechnology involved in treating urinary tract infections: An overview" covers a potentially interesting and emerging topic related to the urinary tract infecions. In this sense, this remains to be potentially interesting for the Nanomaterials readers. I regard the main point of this paper as highly attractive as well as the results are clearly presented. The text does not contain any major errors, therefore I have some minor comments and recommendations:

1. There is a need to provide slightly more expanded introduction shortly
mentioning/describing impact of urinary tract infections on modern healthcare regarding its pharmacoeconomic and costs.

2. The figure summarizing and clarifying the results should be added.

3. Following references should be added and properly cited within the main text:

- Mela A, Poniatowski ŁA, Drop B, Furtak-Niczyporuk M, Jaroszyński J, Wrona W, Staniszewska A, Dąbrowski J, Czajka A, Jagielska B, Wojciechowska M, Niewada M. Overview and Analysis of the Cost of Drug Programs in Poland: Public Payer Expenditures and Coverage of Cancer and Non-Neoplastic Diseases Related Drug Therapies from 2015-2018 Years. Front Pharmacol. 2020 Aug 14;11:1123. doi: 10.3389/fphar.2020.01123.

-Foxman B. The epidemiology of urinary tract infection. Nat Rev Urol. 2010 Dec;7(12):653-60. doi: 10.1038/nrurol.2010.190.

- Mela A, Rdzanek E, Poniatowski ŁA, Jaroszyński J, Furtak-Niczyporuk M, Gałązka-Sobotka M, Olejniczak D, Niewada M, Staniszewska A. Economic Costs of Cardiovascular Diseases in Poland Estimates for 2015-2017 Years. Front Pharmacol. 2020 Sep 8;11:1231. doi: 10.3389/fphar.2020.01231.

- Chenoweth C, Saint S. Preventing catheter-associated urinary tract infections in the intensive care unit. Crit Care Clin. 2013 Jan;29(1):19-32. doi: 10.1016/j.ccc.2012.10.005.

4. In some places the use of English could be improved on.

Completing this gaps will have an impact on the understanding the aim of the study and, from my point of view, is absolutely necessary.

Author Response

Comments and Suggestions for Authors

The by Crintea et al. "Nanotechnology involved in treating urinary tract infections: An overview" covers a potentially interesting and emerging topic related to the urinary tract infecions. In this sense, this remains to be potentially interesting for the Nanomaterials readers. I regard the main point of this paper as highly attractive as well as the results are clearly presented. The text does not contain any major errors, therefore I have some minor comments and recommendations:

  1. There is a need to provide slightly more expanded introduction shortly mentioning/describing impact of urinary tract infections on modern healthcare regarding its pharmacoeconomic and costs.

All that you requested is implemented in the introduction of the manuscript. It is marked in red/blue with track-changes.

  1. The figure summarizing and clarifying the results should be added.

The figure was inserted (fig. 6) and a table was also included.

  1. Following references should be added and properly cited within the main text:

- Mela A, Poniatowski ŁA, Drop B, Furtak-Niczyporuk M, Jaroszyński J, Wrona W, Staniszewska A, Dąbrowski J, Czajka A, Jagielska B, Wojciechowska M, Niewada M. Overview and Analysis of the Cost of Drug Programs in Poland: Public Payer Expenditures and Coverage of Cancer and Non-Neoplastic Diseases Related Drug Therapies from 2015-2018 Years. Front Pharmacol. 2020 Aug 14;11:1123. doi: 10.3389/fphar.2020.01123.

-Foxman B. The epidemiology of urinary tract infection. Nat Rev Urol. 2010 Dec;7(12):653-60. doi: 10.1038/nrurol.2010.190.

- Mela A, Rdzanek E, Poniatowski ŁA, Jaroszyński J, Furtak-Niczyporuk M, Gałązka-Sobotka M, Olejniczak D, Niewada M, Staniszewska A. Economic Costs of Cardiovascular Diseases in Poland Estimates for 2015-2017 Years. Front Pharmacol. 2020 Sep 8;11:1231. doi: 10.3389/fphar.2020.01231.

- Chenoweth C, Saint S. Preventing catheter-associated urinary tract infections in the intensive care unit. Crit Care Clin. 2013 Jan;29(1):19-32. doi: 10.1016/j.ccc.2012.10.005.

The articles of Mela were not mentioned because we did not find in the text appropriate content. But we included Foxman and Chenoweth and also many other references (Stemler and Mysorekar, 2014; Warzecha et al., 2021, etc). We also took data from ECDC and ENSH (ECD - European Centre for Disease Control and ENSH - European Network for Safer Healthcare (https://www.eusaferhealthcare.eu/wp-content/uploads/Increasing-adherence-to-CAUTI-guidelines-recommendations-from-existing-evidence.pdf)

  1. In some places the use of English could be improved on.

We performed another language check.

Completing this gaps will have an impact on the understanding the aim of the study and, from my point of view, is absolutely necessary.

Reviewer 2 Report

Please rephrase the following sentence, it is too long "Although UTIs overall are highly preventable health issues, the recourse to antibiotics as drug treatments for these infections is a worryingly spread approach, especially through the clinicians from least developed countries or even those working in developing nations..."

Also, consider writing a different abstract as it does not quite align with the title of the manuscript.

I dont think lines 105-139 are absolutely necessary for the target audience and can be condensed. This is also true for lines 49-67, it all seems excessive and unnecessary. Moreover, only part from 4. Nanotechnology used as a diagnostic method and in antimicrobial treatment is aligned with the topic of the review, while the previous parts and introductory part lack flow. Part 4. should be expanded with current diagnostic methods, and the manuscript may be better organised keeping just parts from 4. onward and explaining current diagnostic and treatment approach in more scientific (mechanism of action and resistance) and less popular (history) manner.

In conclusion, everything except the last paragraph should be moved to other parts of the manuscript as it more resembles discussion.

Author Response

Thank you very much for reviewing our manuscript! You have the responses below.

Comments and Suggestions for Authors

Please rephrase the following sentence, it is too long "Although UTIs overall are highly preventable health issues, the recourse to antibiotics as drug treatments for these infections is a worryingly spread approach, especially through the clinicians from least developed countries or even those working in developing nations..."

It was rephrased. It is marked in red with track-changes.

Also, consider writing a different abstract as it does not quite align with the title of the manuscript.

The abstract was also re-written. It is marked in red with track-changes.

I dont think lines 105-139 are absolutely necessary for the target audience and can be condensed. This is also true for lines 49-67, it all seems excessive and unnecessary. Moreover, only part from 4. Nanotechnology used as a diagnostic method and in antimicrobial treatment is aligned with the topic of the review, while the previous parts and introductory part lack flow. Part 4. should be expanded with current diagnostic methods, and the manuscript may be better organised keeping just parts from 4. onward and explaining current diagnostic and treatment approach in more scientific (mechanism of action and resistance) and less popular (history) manner.

All requested is done. It is marked with track-changes.

In conclusion, everything except the last paragraph should be moved to other parts of the manuscript as it more resembles discussion.

All requested is done.
